# Reinforcing Linear Low-Density Polyethylene with Surfactant-Treated Microfibrillated Cellulose

**DOI:** 10.3390/polym11030441

**Published:** 2019-03-06

**Authors:** Guangzhao Wang, Xiaohui Yang, Weihong Wang

**Affiliations:** Key Lab of Bio-based Material Science Technology of Education Ministry, Northeast Forestry University, Harbin 150040, China; guangzhaowang1@163.com (G.W.); xiaohuiyang90@163.com (X.Y.)

**Keywords:** microfibrillated cellulose, surfactant, linear low-density polyethylene, composite, properties

## Abstract

Due to its excellent mechanical properties and reinforcement abilities, cellulose has become a promising candidate for developing nanocomposites. However, cellulose agglomeration is an issue that must be solved. In this study, we treated microfibrillated cellulose (MFC) with a mixture of the non-ionic surfactants Span80 and Tween80 (ratio of 1:1) in order to prevent the intermolecular hydrogen bond aggregation of MFC during the process of MFC drying. We used a conical twin-screw extruder to melt compounds for the surfactant-treated MFC and powdered LLDPE. Furthermore, the extruded mixture was hot-pressed into a film, and we also tested the properties of the composite film. We can conclude that there was no agglomeration in the composite film according to microscopic observations and light transmittance test results. Furthermore, the dispersion of the surfactant-treated MFC (STMFC) was uniform until the STMFC filler increased to 10 wt%. The mechanical test results show that when the content of STMFC filler was 10 wt%, the mechanical properties of the composite were optimal. Compared to LLDPE, the STMFC/LLDPE composite film had an increase of 41.03% in tensile strength and an increase of 106.35% in Young’s modulus. Under this system, the DSC results show that the melting point of LLDPE increased from 125 to 131 °C. X-ray diffraction (XRD) results showed that the addition of STMFC did not change the crystallinity of the STMFC/LDPE composites, although the crystallite size increased.

## 1. Introduction

Cellulose can be obtained from several natural sources, such as cotton, sugarcane and wood. It is the most abundant biodegradable polymer material found on the Earth [1,2,3,4,5,6]. The chemical structure of cellulose is formed by D-glucose units that are connected by β-1,4-glycosidic linkages. The –OH groups present at the C_2_, C_3_ and C_6_ atoms of each β-d-glucopyranose unit of a cellulose chain are the most susceptible active sites [7,8,9]. Cellulose has several attractive properties, such as a high modulus, high strength, biodegradability, renewability and good biocompatibility. Moreover, it exhibits good mechanical properties. Its modulus of elasticity is approximately 150 GPa and the tensile strength is approximately 10 GPa [10,11,12,13]. These advantages can be employed to produce polymer composite materials with special functions [14,15].

However, the cellulose surface contains many hydroxyl groups, which results in poor compatibility between the hydrophilic cellulose fibers and the hydrophobic matrix [16]. More importantly, cellulose agglomerates easily by forming hydrogen bonds. Thus, it is important to treat the surface of natural fiber fillers in order to improve the dispersion in a polymer matrix and increase the filler/matrix interfacial interaction. This can allow for greater stress transfer through a strong interface from the reinforcing element to the matrix [17,18]. In order to improve the dispersibility, there are also many methods for improving the matrix/filler interaction and interface, including coupling agent treatment, nanoparticle coating, surfactant treatment and block copolymer molecules through micellization and interaction with the hydrophilic particle surface/hydrophobic organic phase polymer matrix [19,20,21].

Polyolefins, such as polyethylene (PE) and polypropylene (PP), have been widely used in the industry and commerce fields. Among them, linear low density polyethylene (LLDPE) has certain tensile strength and good impact resistance. The crystallinity of LLDPE is also low, which makes LLDPE highly transparent. However, LLDPE is a nonpolar polymer with an inert surface and low surface energy, which limits its application [22]. Therefore, we need to add fillers to change its properties in the preparation of LLDPE composite materials. Due to its environmental protection, renewability and excellent mechanical properties, cellulose has become a promising candidate for developing nanocomposites. The combination of cellulose fibers and LLDPE can improve the strength and polarity of LLDPE to a certain extent and help to prolong the service life of products. Therefore, it is important to determine how to uniformly disperse cellulose into the nonpolar polyolefin matrix and form a good interface in order to improve the reinforcement effect of cellulose [14].

The interfacial compatibility between the cellulose fibers and polymers can be improved by surface modification and interfacial regulation. We usually use methods, such as acetylation, silane coupling and surfactant treatment [23,24,25,26]. In previous studies, Wang et al. [27] adsorbed ethylene acrylate oligomers on the surface of the microfibrillated cellulose (MFC) to enhance PP and PE. The experimental results showed that the dispersion performance of MFC improved but there was no significant mechanical improvement. Mariano et al. [28] improved the dispersion of MFC in PLA by adsorbing a poly(L-lactide)-based pre-polymer on the MFC surface but this reduced the tensile strength of the composite. Bondeson and Oksman et al. [29] used anionic surfactants to treat cellulose whiskers and subsequently filled the PLA matrix. The tensile strength and the elongation at break of the composite was improved compared to its unreinforced counterpart. Iwamoto et al. [30] used polyoxyethylene (10) nonylphenyl ether to coat MFC. The results showed that the injection molded PP (80 wt%)/surface-coated MFC (10 wt%) composites had an increase of 45% in Young’s modulus and an increase of 50% in yield strength compared to the neat PP. Emami et al. [31] used ethylene oxide–propylene oxide block copolymer surfactants to treat cellulose, which increased the storage moduli of the cellulose/epoxy composites by approximately 77% compared to that of the untreated epoxy matrix at high temperatures. Moreover, the glass-rubbery transition temperature of the treated cellulose/epoxy composite increased by approximately 10 °C. Based on these encouraging results, surface coating may be a promising method for improving the dispersibility of MFC in a thermoplastic matrix, which enhanced the mechanical properties of the composite.

Based on these encouraging results, surfactants should be able to easily cover MFC to prevent the intermolecular hydrogen bond aggregation of MFC during the process of MFC drying. The surface coating may be promising for improving the dispersibility of and compatibility between the MFC in the thermoplastic matrix, thus resulting in enhanced mechanical properties of the composite. Span80 and Tween80 are nonionic surfactants, which are usually mixed together to process cellulose. Yang et al. [32] found that the mixture of Span80 and Tween80 at 1:1 ratio was the most stable and MFC was evenly dispersed in the Span80/Tween80 mixture. The primary component of Span80 is sorbitan monooleate, whereas the primary component of Tween80 is polyoxyethylene sorbitan monooleate [33]. Usually, they are used synthetically. Moreover, the hydroxyl groups of the Span80/Tween80 mixture are strongly lipophobic, whereas the long chains are strongly hydrophobic [34]. Therefore, Span80 and Tween80 may be effective for dispersing the hydrophilic cellulose in a nonpolar matrix.

In this study, a mixture of Span80 and Tween80 (S/T) at 1:1 ratio was developed to treat MFC. MFC/LLDPE composite was prepared by an extrusion and hot-pressing process. The mechanical properties, microstructures, crystallinities and interfacial compatibilities of the composites were studied to examine the effect of cellulose dispersion on the enhancement effects in polymers.

## 2. Materials and Methods

### 2.1. Materials

MFC (diameter: 5.00 nm–0.01 μm, length: 400–600 μm; water content: 75%) was purchased from Daicel FineChem Ltd. (Tokyo, Japan). Linear low-density polyethylene (LLDPE) (melt flow index: 0.7 g/10 min at 190 °C, density: 0.923 g/cm^3^) was purchased from the Daqing Petrochemical Company (Daqing, China). Maleic anhydride graft polyethylene (MAPE) (grafting percentage of 0.9%) was obtained from Shanghai Sunny New Technology Development Co., Ltd. (Shanghai, China) Furthermore, Span80, Tween80 (structural formula are shown in Figure 1 and Figure 2), xylene and ethanol were purchased from Zhiyuan Chemical Reagent Co, Ltd. (Tianjin, China).

### 2.2. Treat MFC with Surfactant

MFC with 25% solid content and deionized water were weighed and mixed to give a ratio of 1:99 (the deionized water includes MFC water) before being transferred into the beating engine. The beating engine was stirred 3 times for 5 min each time. The prepared 1 wt% MFC suspension was poured into the bottle before being sealed and refrigerated at about 4 °C for reserve. MFC was added into deionized water and mixed well with the 1% MFC suspension. Span80 (S) and Tween80 (T) were mixed in a ratio of 1:1 and stirred at 50 °C for 30 min. This ratio of 1:1 was optimally decided in a previous study in order to ensure its stability and a uniform dispersion of MFC [32]. The 1% MFC suspension was added into the S/T mixture and stirred at 50 °C for 20 min. After this, the mixture was dried in an oven at 50 °C to reduce the water content by 3–5%.

### 2.3. Mix Surfactant-Treated MFC and LLDPE

After drying, the surfactant-treated MFC (STMFC) was dispersed in xylene to form a MFC/xylene suspension. The LLDPE particles were added into the suspension and stirred at 135 °C for 30 min. The STMFC/LLDPE mixture was subsequently dried in a fume hood at room temperature in order to remove and recycle xylene. After this, the dried mixture was washed with 95% ethanol before being dried at 80 °C. This process was repeated four times.

### 2.4. Preparation STMFC/LLDPE Compound

The dry STMFC/LLDPE mixture was slightly pulverized into small particles using a pulverizer (HNEB-115K, TAISITE, Tianjin, China). The small particles were fed into a corotating, intermeshing twin screw extruder (RuiMing, JZSZ-10A, Wuhan, China). The screws were double-threaded with a length of 109.5 mm and had a conical shape with a screw diameter of 14 mm at the beginning and 5 mm at the end. The processing temperature was 140 °C and the screw speed was 30 rpm. The ratios of MFC, LLDPE and MAPE are listed in Table 1. As a control, the row material of MFC (with 75% moisture content) as received was directly mixed with LLDPE in the same ratio and extruded using the above-mentioned process.

### 2.5. Preparation of MFC/LLDPE Composite Films

The STMFC/LLDPE compound was cut into small particles before being compression-molded using a laboratory hot-pressing machine (ZS-406B, Zhuosheng Machinery Equipment Co., Ltd. Dongguan, China). The compound particles were preheated for 5 min at 140 °C and hot-pressed for 3 min at 140 °C and 5 MPa. The melt sheet was immediately cooled down to room temperature at 3 MPa. A film with dimensions 80 × 110 × 0.5 mm was obtained.

### 2.6. Characterization of the STMFC/LLDPE Composites

#### 2.6.1. Fourier Transform Infrared Spectrometry

The FTIR spectra were implemented to detect the surfactant on MFC. Fourier transform infrared (FTIR) spectrometer (Nicolet Magna IR 560, Madison, WI, USA) was used with a scanning range of 4000–400 cm^−1^, a resolution of 4 cm^−1^ and a scanning number of 32. Attenuated Total Reflectance mode was used for FTIR spectrometry. Both untreated and ST covered MFC were detected by FTIR and dried before testing.

#### 2.6.2. Light Transmittance of Composite Film Test

Light transmittance is the ability of light to pass through a medium [35]. It is often used to evaluate the dispersion of a reinforcing material in a polymer matrix. Light transmittance (Tt) is expressed as the ratio of the luminous flux transmitted through the sample to the luminous flux incident on the sample [36]:(1)$$Tt=\frac{T2}{T1}$$
where *T*t is the transmittance, *T*2 is the total transmitted light flux through the sample and *T*1 is the incident light flux.

Transmittance can be used to characterize the dispersion of fibers in the matrix. TU-1901 dual-beam UV–Vis spectrophotometer (Beijing Pu Ke General Instrument Co., Ltd. Beijing, China) was used to test the transmittance of the composite film. The film was cut into a rectangular sample with dimensions of 10 mm × 20 mm and placed on a sample holder. Testing was conducted at a wavelength of 200–800 nm.

#### 2.6.3. Tensile Property Test

The tensile strength of the STMFC/LLDPE composite film was measured with reference to GB/T 1040.2-2006, which is the standard for the Plastics Determination of Tensile Properties (Part 2: Test conditions for molding and extrusion plastics). Tensile testing was carried out by a Universal Mechanics Experimental Machine (RGT-20A, Rigel Co., Ltd. Shenzhen, China). The dumbbell-type specimen (length of 75 mm) was tested at a speed of 5 mm/min piece, which was replicated six times for each group.

#### 2.6.4. Scanning Electron Microscopy

After tensile testing, the fracture surface of the STMFC/LLDPE film was characterized using a scanning electron microscope (Quanta 200, FEI, Hillsboro, OR, USA) at an acceleration voltage of 5 kV. The surfaces of fracture were sputtered with gold before observation.

#### 2.6.5. X-ray Diffraction

The crystallinity of the STMFC/LLDPE film was collected on a D/max2200 X-ray diffractometer (Rigaku, Tokyo, Japan) with the sample exposed to a high-resolution Cu Kα radiation source at room temperature, 40 kV and 30 mA with a scanning speed of 5°/min from 10° to 40°.

LLDPE is a semi-crystalline polymer. The addition of STMFC affected its crystallization behavior and performance. The crystallite size D can be calculated using Scherrer’s formula:(2)$$D=\frac{K\times \lambda }{\beta \times \cos\theta }$$
where D is the crystallite size that is perpendicular to the (hkl) surface (nm); K is a constant; λ is the incident X-ray wavelength, which was 0.15418 nm; β is the full width at half maximum of the diffraction peak, which was determined by the crystal size; and θ is the Bragg angle, whose value was half of the peak horizontal.

The areas of the (110) and (220) crystal peaks were denoted as *S*_110_ and *S*_220_, respectively, and the unshaped area of the other region was denoted as *S*_amorphous_. The crystallinity was calculated using the following formula:(3)$$X\%=\frac{S_{110}+aS_{220}}{S_{110}aS_{220}+bS_{\mathrm{amorphous}}2a}$$
where a = 1.43 and b = 0.69 [37].

#### 2.6.6. Differential Scanning Calorimetry

Differential canning calorimetry (DSC) was performed using a NETZSCH DSC 204 instrument (DSC-Q20, TA Instruments, New Castle, DE, USA) calibrated with indium and zinc standards under a nitrogen atmosphere. The composite film samples were heated from 25 °C to 200 °C at 10 °C/min and maintained at 200 °C for 3 min in order to completely eliminate any traces of heat. The sample was cooled to 40 °C at a rate of 10 °C/min. After this, the samples were quenched to and reheated from 40 °C to 200 °C at a rate of 10 °C/min. The nitrogen flow rate was set to 50 mL/min.

## 3. Results

### 3.1. Chemical Characteristic of STMFC

FRIT was used to characterize the treatment effect of surfactants on MFC. Surfactant-treated and untreated MFC were analyzed using FTIR (Figure 3). When MFC was treated with the surfactant S/T, the asymmetric stretching vibration peak of C-H at 2900 cm^−1^ became stronger compared to the untreated MFC. This is because the surfactants introduced more C-H groups on its long chains as shown in Figure 2. A new stretching vibration peak of CH_2_ appeared at 2853.79 cm^−1^, which was also introduced by the S/T surfactant. Similarly, for the treated MFC, a new strong absorption peak appeared at 1736.43 cm^−1^. This is the characteristic absorption peak of the C=O in S/T [38,39], which indicates that the surfactant combined well with the MFC. The exposed hydrophobic chains of S/T prevented intermolecular hydrogen bond aggregation of the MFC during drying and reinforced the LLDPE. The fluffy MFC benefits the penetration of molten LLDPE. In addition, the compatibility of LLDPE with MFC was improved and the reinforcement between LLDPE and MFC was enhanced.

### 3.2. Dispersion of MFC in LLDPE Matrix

If MFC agglomeration was observed in the composite film, the film’s transmittance would change between different positions. Figure 4 shows the transmittance curves of the different positions of the same sample. It is important to note that these curves are basically the same as each other. The transmittance curves at different locations of the 10 wt% STMFC/LLDPE are almost unchanged, which indicates that the surfactant treatment improved MFC by ensuring that it was uniformly distributed in the LLDPE matrix. However, when the STMFC content increased to 14 wt%, the coincidence of the three curves was lower than that in the 10 wt% STMFC/LLDPE film. Moreover, the 14 wt% STMFC/LLDPE composite showed lower transmittance compared to the 10 wt% content, which indicates that a higher content of STMFC made dispersion challenging.

Figure 5 shows the transparency of the untreated MFC/LLDPE and STMFC/LLDPE films. There are many small granular regions in the untreated MFC/LLDPE film. Moreover, the transparency of the untreated MFC filled film was very low. Although there was no obvious agglomeration in the STMFC/LLDPE film, it was more transparent, which also indicates that STMFC was uniformly dispersed in the LLDPE matrix.

### 3.3. Tensile Properties

With an increase in the STMFC content, the tensile strength and Young’s modulus of the composites increased (Figure 6). When the STMFC content increased to 10 wt%, the tensile strength and Young’s modulus reached their highest values and increased by 41.03% and 106.35%, respectively, compared to those of LLDPE. As discussed in Section 2.3, LLDPE was mixed with xylene to form a solution. MFC dispersed easily in the LLDPE solution, which allowed the combination of MFC and LLDPE. When the surfactants were deposited on the MFC, the intermolecular hydrogen-bond-induced aggregation of MFC prevented the process of drying. The STMFC subsequently combined well with LLDPE during extrusion and had an excellent enhancement effect on the composites. For the untreated MFC/LLDPE, there was no significant difference in the tensile strength for MFC contents from 2 wt% to 14 wt%.

This may have been due to the incompatibility between the MFC and LLDPE, which resulted in a low shear strength. MFC did not work well due to the transmission of stress. However, a stiff MFC significantly increased the Young’s modulus of LLDPE although this was not uniformly dispersed. For the surfactant-treated MFC, adding 6 wt% STMFC significantly improved the tensile strength and the highest value was attained at 10 wt% STMFC filler due to the improved compatibility after surfactant treatment. However, using an excessive amount of MFC resulted in aggregation and reduced the tensile properties. When the filler of STMFC exceeded 10 wt%, the tensile strength and Young’s modulus of the composites decreased due to the uneven distribution of the STMFC (Figure 4 and Figure 5). For the untreated MFC, the tensile strength improved slightly. Young’s modulus significantly improved but this improvement was less than that in the STMFC/LLDPE composites. Figure 7 shows the stress–strain curves for the STMFC/LLDPE composite. The results show that the maximum stress of composites gradually increased with an increase in the STMFC content but the elongation at break gradually decreased. The trend is the same as that of the mechanical properties in Figure 6. This is because STMFC has good mechanical strength and high modulus and thus, STMFC can reinforce the STMFC/LLDPE composites when it was evenly dispersed in the LLDPE matrix.

### 3.4. Microstructure of the MFC/LLDPE Composite

It is important to note that STMFC showed no obvious agglomeration in the LLDPE matrix. Moreover, it exhibited a certain amount of twining that linked it with LLDPE (Figure 8d–f). Furthermore, the interface between the STMFC and LLDPE matrix improved considerably. Untreated MFC formed large agglomerates in the LLDPE matrix and was stacked. With increased MFC content, the agglomeration was more significant (Figure 8a–c). This led to a stress concentration in the composite, resulting in a decrease in the mechanical properties.

With an increase in the STMFC content, STMFC in the LLDPE matrix was more densely distributed but this distribution was still uniform even at 10% (Figure 8e). It is important to note that the STMFC and LLDPE matrix were closely integrated. Moreover, the mechanical test results showed that a 10 wt% STMFC content was optimal. When the content increased to 14 wt%, STMFC showed partial agglomeration and formed block agglomerates, which led to a decrease in the mechanical properties of the composite.

### 3.5. Crystal Structure of MFC

As shown in Figure 9, 2θ peaks appeared at 20.94° (110) and 23.1° (200), which are the characteristic diffraction peaks of low-density polyethylene [40]. Furthermore, the addition of STMFC did not change the crystal morphology of LLDPE and the crystal shape of the STMFC/LLDPE composite remained orthogonal. Using the X-ray diffraction (XRD) data, the microcrystal size and crystal cell parameters of the sample were calculated and are listed in Table 2.

The crystallite size of the LLDPE composites increased with the addition of STMFC. However, the degree of crystallinity remained nearly unchanged for two primary reasons. The STMFC acted as a nucleating agent, which increased the crystallite sizes of the STMFC/LDPE composites and improved the crystallinity of the STMFC/LLDPE composites. This result was also found by Zahra Emmi [31]. However, the addition of MFC destroyed the regularization of the LLDPE macromolecular chains and restricted the movement of the LLDPE composite molecular chains, leading to a reduction in the crystallinity. Therefore, the combination of these two factors resulted in insignificant changes in the crystallinity of the STMFC/LLDPE composites. The physical properties of the MFC/LLDPE composites largely depended on the crystalline nature of the LLDPE.

### 3.6. Thermal Stability of STMFC/LLDPE Composite Film

Figure 10 shows the DSC melting curves of the STMFC composite films with different contents. As the temperature increased, the thermal motion energy of the LDPE molecule increased, resulting in the destruction of the crystals and the state of the substance changed from a crystalline to a liquid state. Therefore, a distinct exothermic peak appeared. As shown in Figure 10, the melting peaks of the STMFC/LLDPE films shifted to higher temperatures with increasing STMFC contents.

The melting point of the STMFC/LLDPE composite was significantly higher than that for the neat LLDPE. The melting point of the neat LLDPE was 125 °C while the melting point of the 10% STMFC/LLDPE composite was 131 °C. Thus, the melting point of the composite increased by 6 °C. The introduction of STMFC improved the thermal stability of the LLDPE. Moreover, the crystallite size of the LLDPE composites increased with the addition of STMFC according to the XRD test results (Table 2). This may be responsible for the improved thermal stability of the composite compared to LLDPE.

## 4. Conclusions

In this study, MFC was treated with surfactant mixtures of Span80 and Tween80 before being reinforced with LLDPE. When STMFC was added to a level of 10 wt%, the tensile strength and Young’s modulus of the STMFC/LLDPE composite film reached their highest values and increased by 41.03% and 106.35%, respectively, compared to neat LLDPE. When STMFC was >10 wt%, the tensile strength and Young’s modulus of the composites partially decreased, and the melting point of the STMFC/LLDPE composite was significantly higher than that of the neat LLDPE. It is important to note that the transmittance shows that the dispersion of STMFC in LLDPE was greatly improved compared to the untreated MFC. The addition of STMFC enlarged the size of crystallites but did not change the crystallinity of the STMFC/LLDPE composites. Furthermore, the melting point of the STMFC/LLDPE composites was improved. With the help of the Span80 and Tween80 surfactants, the process used in this study effectively reduced the agglomeration of MFC in the LLDPE matrix. Moreover, STMFC significantly improved the tensile properties and thermal stability of LLDPE.

## Figures and Tables

**Figure 1 polymers-11-00441-f001:**
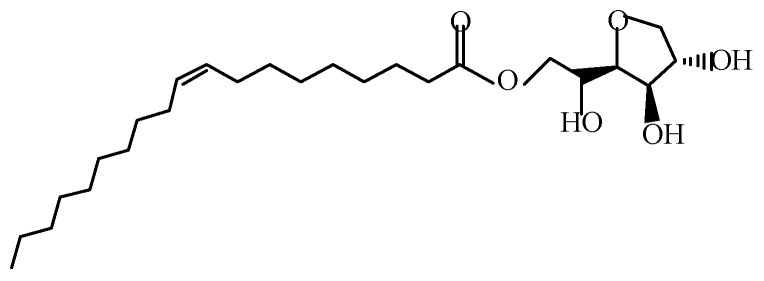
Structural formula of Span80.

**Figure 2 polymers-11-00441-f002:**
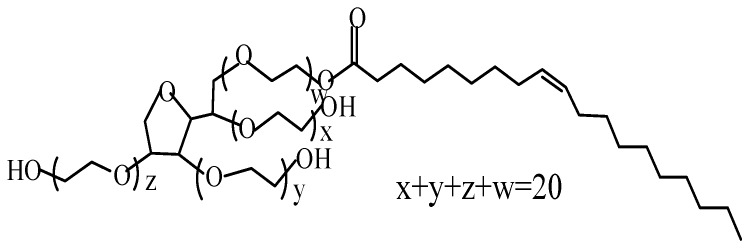
Structural formula of Tween80.

**Figure 3 polymers-11-00441-f003:**
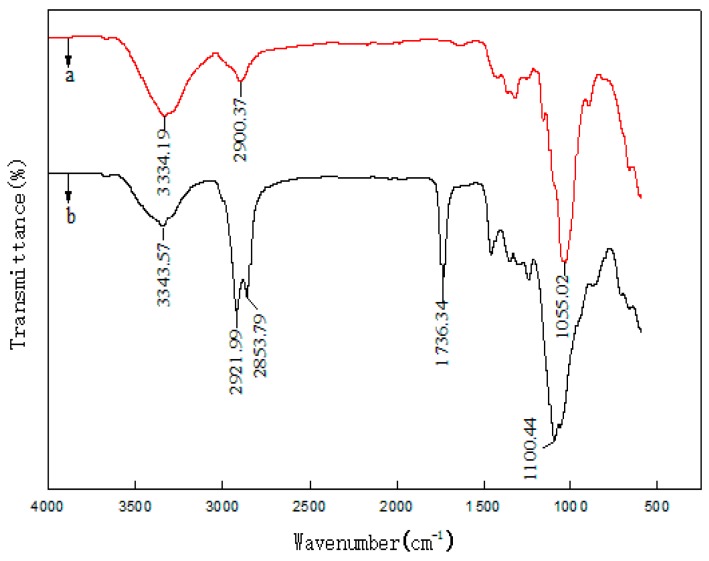
FTIR spectra of untreated (**a**) MFC and (**b**) surfactant-treated MFC.

**Figure 4 polymers-11-00441-f004:**
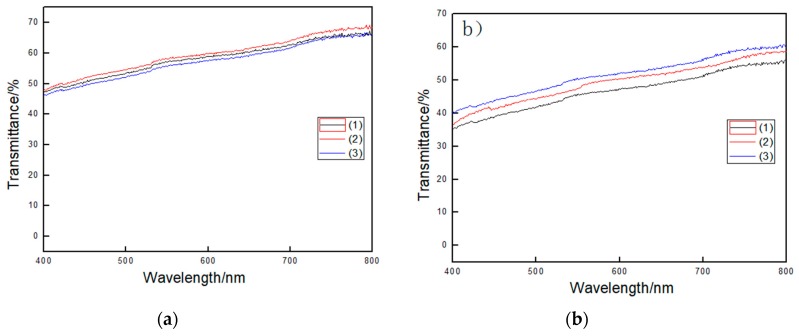
Transmittance at three different positions on the same sample: (**a**) 10 wt% STMFC/LLDPE and (**b**) 14 wt% STMFC/LLDPE.

**Figure 5 polymers-11-00441-f005:**
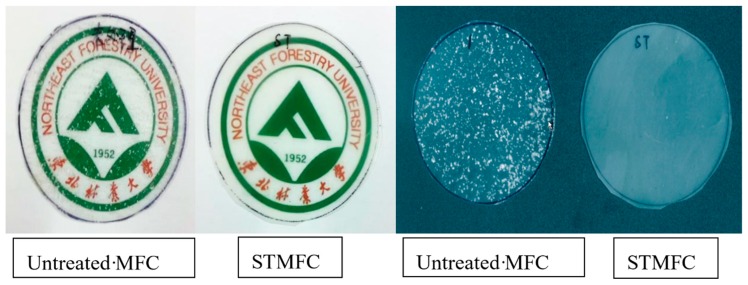
Scanning images of LLDPE composite films enhanced by 10 wt% MFC with different processing methods.

**Figure 6 polymers-11-00441-f006:**
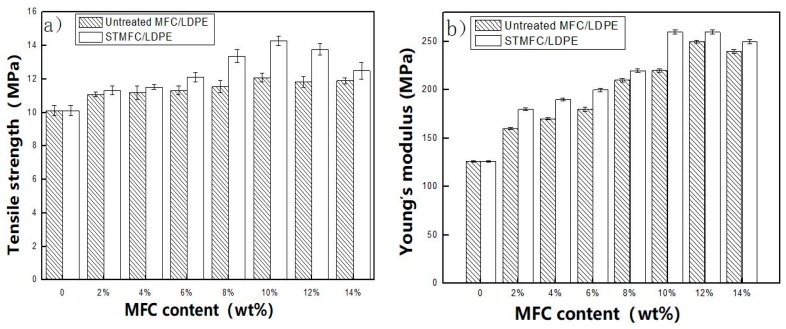
Mechanical properties of surfactant-treated and untreated microfibrillated cellulose-reinforced LLDPE composites with different MFC contents (wt%): (**a**) tensile strength and (**b**) Young’s modulus.

**Figure 7 polymers-11-00441-f007:**
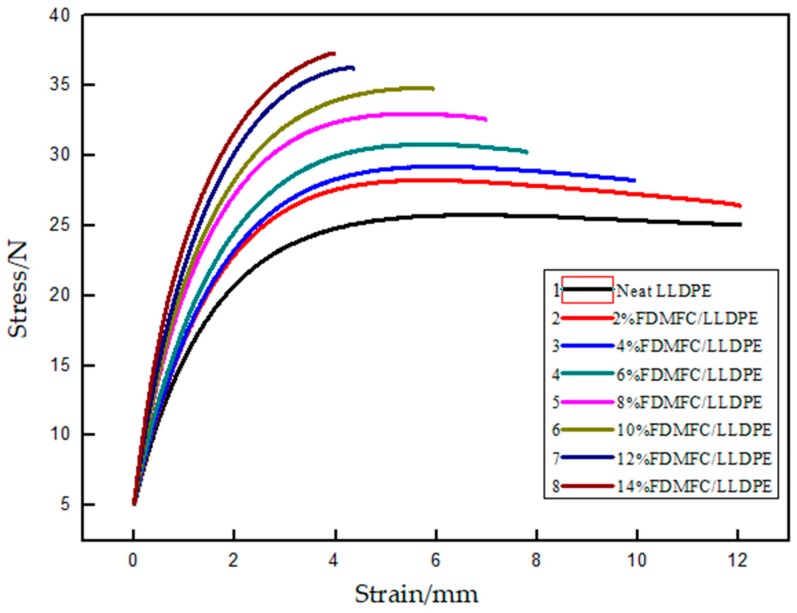
Stress–strain curves of STMFC/LLDPE composites with different MFC contents.

**Figure 8 polymers-11-00441-f008:**
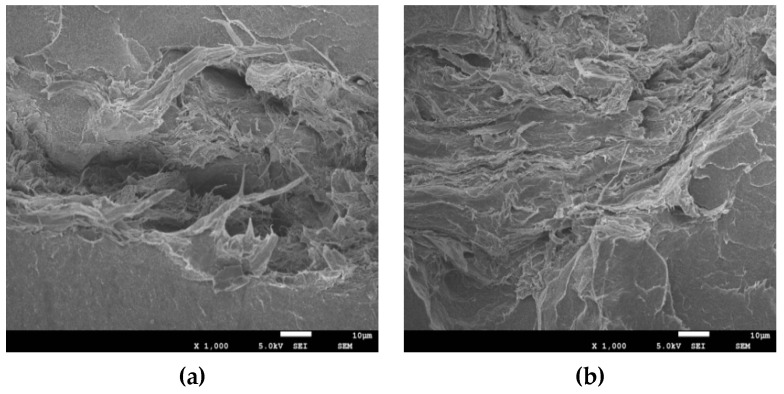
SEM images of untreated MFC/LLDPE and STMFC/LLDPE composites. Untreated MFC: (**a**) 4 wt%, (**b**) 10 wt% and (**c**) 14 wt%. STMFC: (**d**) 4 wt%, (**e**) 10 wt% and (**f**) 14 wt%.

**Figure 9 polymers-11-00441-f009:**
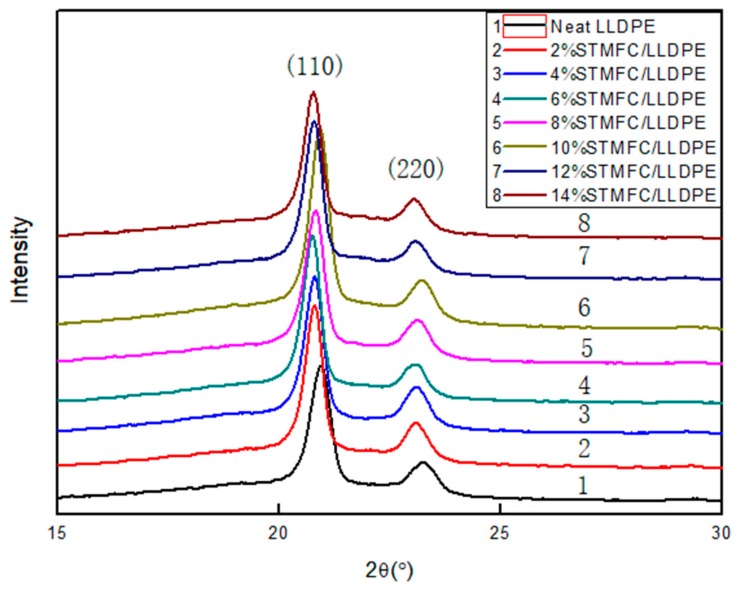
XRD curves of STMFC/LLDPE composite films with different contents.

**Figure 10 polymers-11-00441-f010:**
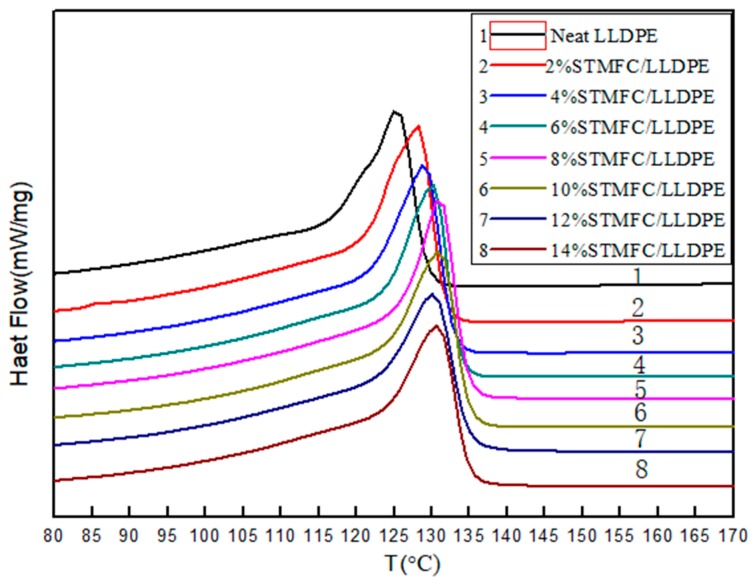
DSC melting curves of STMFC/LLDPE composites with different contents.

**Table 1 polymers-11-00441-t001:** Formulation of MFC/LLDPE mixed extrusion experiments.

MFC/wt%	MAPE/wt%	LLDPE/wt%
0	5	95
2	5	93
4	5	91
6	5	89
8	5	87
10	5	85
12	5	83
14	5	81

**Table 2 polymers-11-00441-t002:** Crystal parameters of the STMFC/LLDPE composite reinforced with different contents.

STMFC (wt%)	PEAK 2θ (°)		Crystallite Size (nm)		FWHMβ (nm)		Degree of Crystallinity X (%)
	110	220	110	220	110	220	
0	20.94	23.24	12.06	8.96	0.65	0.89	77.80
2	20.80	23.06	13.39	10.57	0.59	0.76	77.32
4	20.80	23.08	13.20	10.58	0.61	0.78	79.72
6	20.76	23.10	13.92	9.93	0.58	0.81	78.01
8	20.82	23.12	12.51	9.72	0.64	0.83	78.12
10	20.78	23.10	13.31	11.39	0.60	0.70	79.01
12	20.78	23.06	13.31	9.04	0.60	0.86	75.91
14	20.76	23.06	13.20	10.45	0.61	0.77	77.91

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
