# Peer review of "Reinforcing Linear Low-Density Polyethylene with Surfactant-Treated Microfibrillated Cellulose"

_polymers, 2019, doi:10.3390/polym11030441_

Round 1
Reviewer 1 Report
It is a well-written and clear to understand. However, to be published, the authors should make more scientific discussions.
1. The authors described “Span80 and Tween80 might be beneficial for dispersing the hydrophilic cellulose in nonpolar matrix.” “a mixture of surfactant Span80 and Tween80 was developed to treat MFC” (L. 61-63). Why the authors decided 1:1 ratio? The authors should add different ratio of Span80 to Tween80, not only 1:1.
2. The authors should add more discussions about FTIR results. The surfactant-treated MFC data with different ratio of Span80 and Tween80 should be beneficial.
3. Fig.4: Strange behavior was observed between ~420 - ~550 nm. Why?
4. Fig.5: What percentage of MFC was added?
5. Fig.6: Typical S-S curves should be added. Different conditions (ratio of Span80 and Tween80) may be effective for improving mechanical properties
6. “Crystal structure of MFC” (L 215): The authors described about the crystal structures of LLDPE, not MFC. Were there no cellulose-derived peaks? Why the degree of peak 2Ɵ was different between 0% and others?
Author Response
Response to Reviewer 1 Comments
Point 1: The authors described “Span80 and Tween80 might be beneficial for dispersing the hydrophilic cellulose in nonpolar matrix.” “a mixture of surfactant Span80 and Tween80 was developed to treat MFC” (L. 61-63). Why the authors decided 1:1 ratio? The authors should add different ratio of Span80 to Tween80, not only 1:1.
Response 1: Thanks for your suggestions.
We did study different ratio of Span80 to Tween80 in previous work. First we concern the stability of the mixture of Span80 and Tween80 emulsion at different ration, and second we concern the dispersion of MFC in this mixture. Thus the ratio of Span80 and Tween80 at 1:0.3, 1:0.5, 1:1, 1:2, 1:3 was designed. The results showed that mixture at 1:1 ratio was the stablest and MFC was evenly dispersed in Span80/Tween80 mixture at this ratio.
Fig. 1. (a) Zeta potential and (b) complex viscosity (η*) of MFC/surfactant emulsion.
Fig. 2. Photographs of surfactant-coated MFC suspended in xylene (b) at different S/T ratios (MFC: surfactant mixture = 1:8) and (a) at different MFC/surfactant mixture ratios (S:T=1:1).
This work was mainly introduced in Xiaohui Yang’s thesis She is contributing the paper. The different ratio of Span80 and Tween80 has been discussed in another article and she is contributing the paper. So I didn't add this part.
Point 2: The authors should add more discussions about FTIR results. The surfactant-treated MFC data with different ratio of Span80 and Tween80 should be beneficial.
Response 2: Thanks for your suggestions. We added some more discussion about FTIR results in line 188, line191-193 and line197-198.
Point 3: Fig.4: Strange behavior was observed between ~420 - ~550 nm. Why?
Response 3: Sorry. This was resulted from instrumental error. Unfortunately, we can only retest it in March due to the long winter vacation. We can’t get the result within the time limit for modification. The purpose of this figure is to compare the relative transmittance of three samples. Thus Figure 4 is still used in this article before we get new curves.
Point 4: Fig.5: What percentage of MFC was added?
Response 4: Sorry for my poor description. 10wt% MFC was added.
It has been added in Figure 5 in the new version.
Point 5: Fig.6: Typical S-S curves should be added. Different conditions (ratio of Span80 and Tween80) may be effective for improving mechanical properties.
Response 5: Thanks for your suggestions. Fig.6 was built according to the STMFC/LLDPE composite’s Typical S-S curves. In each group we tested 5-7 samples. There are too many curves for 8 groups. Therefore, we did not add these Typical S-S curves in the article. We have listed some S-S curves below. In addition, as explained earlier, the effect of different conditions (ratio of Span80 and Tween80) was not discussed here.
S-S curves of 4wt% STMFC/LLDPE composite
S-S curves of 10wt% STMFC/LLDPE composite
S-S curves of 14wt% STMFC/LLDPE composite
Point 6: “Crystal structure of MFC” (L 215): The authors described about the crystal structures of LLDPE, not MFC. Were there no cellulose-derived peaks? Why the degree of peak 2Ɵ was different between 0% and others?
Response 6: Thanks for your suggestions. I am sorry for my poor description.
(1) Cellulose-derived peaks are 2Ɵ=22°and2Ɵ=16°, but because the main content of MFC/LLDPE composite film is LLDPE, the content of MFC is lower than that of LLDPE. So the characteristic peaks of LLDPE appeared in the XRD test results, there were not cellulose-derived peaks.
(2) In fact, Table 2 shown the XRD curves of MFC/LLDPE composites with different MFC contents were very similar. And the addition of MFC can act as nucleating agent; this may be the reason the degree of peak 2Ɵ was different.

Reviewer 2 Report
The authors report on a versatile and scalable method for the fabrication of Low-Density Polyethylene/ Cellulose composites via melt mixing process. The work is in general quite innovative and the experimentation that has been performed is adequate for the publication of this research work.
In addition, the paper in general is well-written and contains enough experimental results to support its main findings. However, a few things should be considered.
Line 12: in abstract the authors should mention that the surfactants used as commercial ones and should mention their polarity characteristics (are they anionic, cationic or non-ionic surfactants and what is the expected chemical interaction of the surfactant with the surface of cellulose allowing a high quality of dispersion).
Line 17: “usage” is not propel language. The author should use the term “filler” or “inclusion”. In addition the 10% should be defined in introduction (is it wt% or phr?; phr is used normally for micro-scaled fillers reinforcing a polymer matrix, however wt% is also acceptable).
In abstract, it should be mentioned which surfactant worked better and which system has brought the results e.g Young’s modules, crystallinity, etc. referred.
Line 30: “large modulus” is not formal language.
Line 33: not “composite materials” but polymer composite materials. Is it possible to fabricate metallic or ceramic matric/ cellulose composite materials. This should be considered throughout the whole manuscript; “polymer composites” and not simply “composites”:
Line 34-38: the paragraph is too short and does not contain the required info for the reader to understand why it is important to treat the surface of natural fiber fillers e.g i) for improving both the dispersion in a polymer matrix, as well as ii) via “compatibillisation” increasing the filler/ matrix interaction and the interfacial interaction and strength allowing higher stress transfer through a strong interface from the matrix to the reinforcing element (https://www.sciencedirect.com/science/article/pii/S0261306914001009; https://www.sciencedirect.com/science/article/pii/S0008622314002103). In addition, all the methodologies to improve matrix/ filler (micro- or nanofiller) interaction and interface allowing also an effective dispersion should be mentioned: i) coupling agents; ii) hierarchical nanoparticle coatings on micro scale reinforcements (https://www.sciencedirect.com/science/article/pii/S0266353818302574; https://pubs.rsc.org/ru/content/articlelanding/2016/ra/c6ra09800b/unauth#!divAbstract https://www.sciencedirect.com/science/article/pii/S2452213916300183), and iii) surfactants of both small molecules and block copolymer molecules through micellization and interaction with hydrophilic particle surface/ hydrophobic organic phase polymer matrix (https://www.hindawi.com/journals/jnm/2017/3852310/abs/).
Line 38-55: the authors very nicely describe all specific modification that have been used for cellulose both to create a good dispersion and interfacial interaction with polymer matrices, as well as to create micro cellulose filler.
Line 62: not “beneficial” but “effective”.
Line 126: “sprayed” or “sputtered” with gold.
Line 173: Figure 5 should be improved to bring all the graph at the same line (should use alignment) and also the photos should not be stretched in the Y-direction.
Part 3.3. of SEM investigation: the images is shown that have been captured using 5KV acceleration voltage, however in the experimental part the authors have mentioned that have taken the images at 15 KV. These are very serious mistakes and show that the authors do not know what they have done and what has been the experimental procedures they have followed in their paper.
For the DSC part: Is it possible the authors as have been done in other studies (https://www.sciencedirect.com/science/article/pii/S0032386112006660; https://www.sciencedirect.com/science/article/pii/S0040603117302101) to calculate via DSC the amount of crystallinity for the LDPE/ MFC composites they have fabricated.
The authors should try to improve the Abstract, the Introduction part and significant improvement is needed for the Experimental and Results & Discussion part. Moreover, the language used throughout the manuscript should be significantly improved towards more formal language.
The findings of the paper are sufficiently novel to warrant its publication, however, after including and considering all the major changes proposed.
Author Response
Response to Reviewer 2 Comments
Point 1: Line 12: in abstract the authors should mention that the surfactants used as commercial ones and should mention their polarity characteristics (are they anionic, cationic or non-ionic surfactants and what is the expected chemical interaction of the surfactant with the surface of cellulose allowing a high quality of dispersion).
Response 1: Thanks for your suggestions. According to your opinions, we have made a new revision in line11-14, line18-24 and line26-27.
Point 2: Line 17: “usage” is not propel language. The author should use the term “filler” or “inclusion”. In addition the 10% should be defined in introduction (is it wt% or phr?; phr is used normally for micro-scaled fillers reinforcing a polymer matrix, however wt% is also acceptable).
Response 2: Thanks for your suggestions. We have replaced “usage” with “filler” in line18, line248, line251. In addition, we have added “wt%” behind the percentage of content.
Point 3: In abstract, it should be mentioned which surfactant worked better and which system has brought the results e.g Young’s modules, crystallinity, etc. referred
Response 3: Thanks for your suggestions. We have revised the abstract according reviewers’ suggestions.
Point 4: Line 30: “large modulus” is not formal language.
Response 4: Thanks for your suggestions. I am sorry for my poor description. We have replaced “large modulus” with “high modulus” in line36.
Point 5: Line 33: not “composite materials” but polymer composite materials. Is it possible to fabricate metallic or ceramic matric/ cellulose composite materials. This should be considered throughout the whole manuscript; “polymer composites” and not simply “composites”
Response 5: Thanks for your suggestions. I am sorry for my poor description. We have replaced “composite materials” with “polymer composite materials” in line 40 and other places throughout the whole manuscript.
Point 6: Line 34-38: the paragraph is too short and does not contain the required info for the reader to understand why it is important to treat the surface of natural fiber fillers e.g i) for improving both the dispersion in a polymer matrix, as well as ii) via “compatibillisation” increasing the filler/ matrix interaction and the interfacial interaction and strength allowing higher stress transfer through a strong interface from the matrix to the reinforcing element (https://www.sciencedirect.com/science/article/pii/S0261306914001009; https://www.sciencedirect.com/science/article/pii/S0008622314002103). In addition, all the methodologies to improve matrix/ filler (micro- or nanofiller) interaction and interface allowing also an effective dispersion should be mentioned: i) coupling agents; ii) hierarchical nanoparticle coatings on micro scale reinforcements (https://www.sciencedirect.com/science/article/pii/S0266353818302574; https://pubs.rsc.org/ru/content/articlelanding/2016/ra/c6ra09800b/unauth#!divAbstract https://www.sciencedirect.com/science/article/pii/S2452213916300183), and iii) surfactants of both small molecules and block copolymer molecules through micellization and interaction with hydrophilic particle surface/ hydrophobic organic phase polymer matrix (https://www.hindawi.com/journals/jnm/2017/3852310/abs/).?
Response 6: Thank you very much for your suggestions. We have made major revision about this section in line 50-60.
Point 7: Line 38-55: the authors very nicely describe all specific modification that have been used for cellulose both to create a good dispersion and interfacial interaction with polymer matrices, as well as to create micro cellulose filler.
Response 7: Thanks for your agreement.
Point 8: Line 62: not “beneficial” but “effective”.
Response 8: Thanks for your suggestions. We have replaced “beneficial” with “effective” in line 88.
Point 9: Line 126: “sprayed” or “sputtered” with gold.
Response 9: Thanks for your suggestions. We have replaced “sprayed” with “sputtered” in line 164.
Point 10: Line 173: Figure 5 should be improved to bring all the graph at the same line (should use alignment) and also the photos should not be stretched in the Y-direction.
Response 10: Thanks for your suggestions. According to your suggestion, we have made a new revision about Fig.5.
Point 11: Part 3.3. of SEM investigation: the images is shown that have been captured using 5KV acceleration voltage, however in the experimental part the authors have mentioned that have taken the images at 15 KV. These are very serious mistakes and show that the authors do not know what they have done and what has been the experimental procedures they have followed in their paper.
Response 11: I'm very sorry that my negligence caused these mistakes.
I participated in the whole test process during the test experiment of SEM. I am sure that know what we have done and what has been the experimental procedures we have followed. We have replaced “15KV” with “5KV” in line164.
Point 12: For the DSC part: Is it possible the authors as have been done in other studies (https://www.sciencedirect.com/science/article/pii/S0032386112006660; https://www.sciencedirect.com/science/article/pii/S0040603117302101) to calculate via DSC the amount of crystallinity for the LDPE/ MFC composites they have fabricated.
Response 12: Thanks for your suggestions. I am sorry for my poor description.
We want to analysis the melting point of the STMFC/LLDPE composite by DSC, the crystallinity of STMFC/LLDPE composites has calculated in section 3.5 XRD results. Therefore, we did not calculate via DSC the amount of crystallinity.
Point 13: The authors should try to improve the Abstract, the Introduction part and significant improvement is needed for the Experimental and Results & Discussion part. Moreover, the language used throughout the manuscript should be significantly improved towards more formal language.
Response 13: Thanks for your carefully reviews. We have made major revision and improve the language. The comments from the editor and reviewers help me a lot. We hope the revised manuscript can meet the editor and reviewers’ requirement and can be published in this journal.
Reviewer 3 Report
This work presented by Wang et al. reported cellulose treated with surfactant to reduce the MFC agglomeration, and further blended with LLDPE by using a conical twin-screw extruder and hot pressing machine. Generally, the authors displayed systematically methods and results in the manuscript to conclude the effect of surfactant on cellulose.
However, the greatest deficiency, basically, was the lack of detailed discussion. The authors should add more discussion, especially focusing on the interaction-structure-property relation. The detailed comments are as follows.
1. The abstract should include some conclusion shortly.
2. Introduction section. The authors should indicate why the cellulose has to blend with hydrophobic matrix. Many researches have showed the compatibility with hydrophilic polymers, such as starch, chitosan etc.
3. Line 56. What does the "presumed" mean?
4. Section 2.2. The authors should describe more details about the preparation of 1% MFC suspension in deionized water.
5. Fig 3. Add more discussion about the 2921.99 and 2853.79 cm-1 of STMFC. And authors should explain the spectra changes to greater degree for more information of the newly formed interaction.
6. Line 151-155 should be listed in section 2.6.2.
7. The MFC agglomeration (i.e. the dispersion) should be further observed and detailed described by SEM or TEM.
8. Line 197. Can the authors provide more evidences about the description: the usage of STMFC exceeded 10% caused uneven distribution of STMFC?
9. Fig 7. I can't understand what the arrows mean. Too many arrows were displayed.
10. Line 216-223 should also be listed in Methods section.
Author Response
Response to Reviewer 3 Comments
Point 1: The abstract should include some conclusion shortly.
Response 1: Thanks for your suggestions.We have revised the abstract.
Point 2: Introduction section. The authors should indicate why the cellulose has to blend with hydrophobic matrix. Many researches have showed the compatibility with hydrophilic polymers, such as starch, chitosan etc.
Response 2: Thanks for your suggestions. The reason for blending with hydrophobic matrix has been added in Introduction session in line50-60.
Point 3: Line 56. What does the "presumed" mean?
Response 3: I am sorry for my poor description. We have replaced "presumed" with "considered" in line80.
Point 4: Section 2.2. The authors should describe more details about the preparation of 1% MFC suspension in deionized water.
Response 4: We have introduced the preparation of 1% MFC suspension in deionized water in line 108-111.
Point 5: Fig 3. Add more discussion about the 2921.99 and 2853.79 cm-1 of STMFC. And authors should explain the spectra changes to greater degree for more information of the newly formed interaction.
Response 5: Thanks for your suggestions. We have added more discussion about the FRIT results in line 188, line 191-193 and line 197-198.
Point 6: Line 151-155 should be listed in section 2.6.2.
Response 6: Thanks for your suggestions. We have listed line 151-155 in section 2.6.2 in line145-150.
Point 7: The MFC agglomeration (i.e. the dispersion) should be further observed and detailed described by SEM or TEM.
Response 7: Thanks for your suggestions. For the analysis of MFC dispersion, we analyzed the scanning pictures of the composites and described the MFC dispersion as a whole in section 3.2 Fig.5. And from Fig.5 we can see there are many small granular reunions in the untreated MFC/LLDPE film. Moreover, there was no obvious agglomeration in the STMFC/LLDPE film, which also indicates that STMFC was uniformly dispersed in the LLDPE matrix. In section 3.4 Fig.6, the SEM images of untreated MFC/LLDPE and STMFC/LLDPE composites were further observed the MFC agglomeration.
Point 8: Line 197. Can the authors provide more evidences about the description: the usage of STMFC exceeded 10% caused uneven distribution of STMFC?
Response 8: Thanks for your suggestions. We have listed the evidences at the end of this sentence.
Point 9: Fig 7. I can't understand what the arrows mean. Too many arrows were displayed.
Response 9: Thanks for your suggestions. We used arrows to mark the agglomeration of MFC. According to your opinions, I have deleted the arrows.
Point 10: Line 216-223 should also be listed in Methods section.
Response 10: Thanks for your suggestions. We have listed line 216-223 in Methods section 2.6.5 in line 169-178.

Round 2
Reviewer 1 Report
This paper was returned with little modification. I can’t believe the manuscript has been significantly improved.
Author Response
I apologize for my reply first time. And I have made some new revisions about your comments. For english language and style, we have applied for MDPI's English editing service. I hope you will accept my revisions.
Point 1: The authors described “Span80 and Tween80 might be beneficial for dispersing the hydrophilic cellulose in nonpolar matrix.” “a mixture of surfactant Span80 and Tween80 was developed to treat MFC” (L. 61-63). Why the authors decided 1:1 ratio? The authors should add different ratio of Span80 to Tween80, not only 1:1.
Response 1: Thanks for your suggestions. I am sorry for my poor description.
In previous work, we first studied the stability of the Span80 and Tween80 emulsion and the dispersion of MFC in different ratio of Span80 and Tween80 surfactant mixture. We have mixed different ratio (1:0.3, 1:0.5, 1:1, 1:2, 1:3) of Span80 and Tween80, and we found that the stability of the Span80 and Tween80 emulsion was different when Span80 and Tween80 was mixed by different ratio. We show the stability of the mixture S/T by the level of zeta potential in Fig.1. On the other hand, the dispersion of MFC in different ratio of Span80 and Tween80 surfactant mixture was also different. Through experimental research, the 1:1ratio of Span80 and Tween80 emulsion was the stablest, and MFC can be more evenly dispersed in the 1:1ratio of Span80 and Tween80. So we determined that the ratio of 1:1 is optimal. And the manuscript of this work was mainly introduced by Yang in another article. We have quoted this article in line 82 and line113. (Figure 1 is shown in the attachment.)
Point 2: The authors should add more discussions about FTIR results. The surfactant-treated MFC data with different ratio of Span80 and Tween80 should be beneficial.
Response 2: Thanks for your suggestions. We have added some more discussion about FTIR results in line 194-196 and line200-202.
Point 3: Fig.4: Strange behavior was observed between ~420 - ~550 nm. Why?
Response 3: Thanks for your suggestions. We have retested it with a new machine and replaced Fig.4 with a new one in manuscript.
Point 4: Fig.5: What percentage of MFC was added?
Response 4: Thanks for your suggestions. I am sorry for my poor description.
In Fig.5, Scanning images of LLDPE composite films enhanced by 10wt% MFC with different processing methods. It has been added in Figure 5 in the new revision.
Point 5: Fig.6: Typical S-S curves should be added. Different conditions (ratio of Span80 and Tween80) may be effective for improving mechanical properties.
Response 5: Thanks for your suggestions. I am sorry for my poor description.
We have added new S-S curves of STMFC/LLDPE composites with different MFC contents in Fig.7, and we have given a brief introduction to the content of the pictures in line 263-268.
Point 6: “Crystal structure of MFC” (L 215): The authors described about the crystal structures of LLDPE, not MFC. Were there no cellulose-derived peaks? Why the degree of peak 2Ɵ was different between 0% and others?
Response 6: Thanks for your suggestions. I am sorry for my poor description.
(1) Cellulose-derived peaks are 2Ɵ=22°and2Ɵ=16°, but because the main content of MFC/LLDPE composite film is LLDPE, the content of MFC is lower than that of LLDPE. So the characteristic peaks of LLDPE appeared in the XRD test results, there were not cellulose-derived peaks.
(2) In fact, Table 2 shown the XRD curves of MFC/LLDPE composites with different MFC contents were very similar. And the addition of MFC can act as nucleating agent; this may be the reason the degree of peak 2Ɵ was different.

Reviewer 2 Report
The authors have significantly improved the manuscript in every aspect and can be accepted for publication in its current form.
Author Response
Thanks for your comments! For english language and style, we have applied for MDPI's English editing service.
Reviewer 3 Report
My comments have been responded accordingly. However, I suggest the abstract should be perfected.
Author Response
Thanks for your suggestions. We have made some new revisions to Abstract in manscript, and I hope you can accept my revisions.
Round 3
Reviewer 1 Report
The manuscript has been revised well and suitable for the publication in Polymers now.